# Applying the COM-B behaviour model to understand factors which impact 15–16 year old students' ability to protect themselves against acquirement of Human Papilloma virus (HPV) in Northern Ireland, UK

**Terri Flood**[1]*, **Ciara M. Hughes**[1], **Iseult Wilson**[2], **Marian McLaughlin**[3]

**1** School of Health Sciences, Ulster University, Londonderry, Derry, United Kingdom, **2** School of Nursing and Midwifery, Queen's University Belfast, Belfast, United Kingdom, **3** School of Psychology, Ulster University, Londonderry, Derry, United Kingdom

* t.flood@ulster.ac.uk

**Data Availability Statement:** Data collected from the small focus groups within this study contain

## Abstract

High-risk strains of Human Papillomavirus (HPV) can lead to the development of a number of cancers including cervical, vulvar, penile, anal and oropharyngeal. HPV vaccination programmes offer the HPV vaccine to males and females 12–13 years old in schools throughout the UK. However, knowledge of HPV remains low in post-primary schools. The aim of this study is to capture 15–16 year old students' perceptions regarding the current provision of HPV education, and whether providing HPV education to 15–16 year olds could influence their intention to be vaccinated and/or future sexual health decisions related to HPV. Between 5th November 2021 and 6th May 2022, seven focus groups were conducted with 34 students in post-primary schools in Northern Ireland, United Kingdom. The data was analysed using the COM-B behaviour model to explore the perceived facilitators and barriers impacting students' ability to protect themselves from acquirement of HPV. Students perceived their knowledge of HPV to be poor and supported the addition of comprehensive mandatory HPV education at 15–16 years old when many of them were becoming sexually active. They identified barriers including lack of parental education, school ethos and religion and insufficient education regarding their legal rights to self-consent to HPV vaccination. Students felt that removal of these barriers would lead to safer sexual practices, increased awareness of the importance of HPV screening and increased HPV vaccination uptake. The recommendations provided by students need to be supported by the Education Authority in conjunction with the Department of Health in order to be successfully implemented into the post-primary school curriculum.

## 1. Introduction

In the UK, the impact of sexually transmitted infections (STIs) remains greatest in young people aged from 15–24 years old [1]. By the end of 2022, many STIs had returned to the high

vulnerable populations. The sharing of focus group excerpts would violate the agreement to which the participants consented. Data requests may be e-mailed to the NHS Research Ethics Committee quoting IRAS ID 287358 at surreyborders.rec@hra.nhs.uk.

**Funding:** This work was supported by a grant allocated to Terri Flood, one of the researchers involved in this study. The grant was provided by the College of Radiographers, UK, through the CoRIPs Research Grant fund, grant number 189. The funders had no role in study design, data collection and analysis, decision to publish, or preparation of the manuscript. None of the authors received a salary from any funder.

**Competing interests:** The authors have declared that no competing interests exist.

levels reported in 2019 prior to the coronavirus (COVID-19) pandemic [1]. Genital warts are the most common viral STI diagnosed in the UK and are largely caused by human papilloma-virus (HPV) [2]. As well as genital warts, high risk HPV strains can lead to the development of a number of cancers including cervical, vulvar, penile, anal and oropharyngeal through oral sex, anal sex or vaginal sex [3]. HPV, unlike other common STIs like gonorrhoea, chlamydia, and syphilis, can be prevented through administration of vaccines. HPV vaccination has been introduced into at least 111 countries globally as part of national immunisation programmes [4]. In 2008, the UK introduced HPV vaccination into a post-primary school-based immunisa-tion programme for females aged 12–13 years old [5]; in 2019 the programme was extended to males, 12–13 years old [6]. Specialist public heath trust nurses called Immunisation Nurses (IMNs) deliver this immunisation programme in post-primary schools [7]. IMNs must have minimum education and training standards to qualify as an IMN in the UK [8], with many IMNs also having a post-graduate qualification in public health [9].

Since the introduction of the HPV immunisation programme, cervical cancer rates in the UK have fallen by 87% [10]. This has been partially attributed to high HPV vaccination uptake throughout the UK [10] pre-COVID-19, with 82–84% of females receiving at least one HPV vaccine in the 1st year of post-primary school in 2018–2019 academic school year in England, Scotland and Northern Ireland (84% in Wales in 2019–2020) [11–14]. Despite the majority of legal COVID-19 restrictions ending in March 2022 [15], uptake has remained lower for females than pre-COVID levels with 1st year 1st dose completion rates for the academic year 2021–2022 of 74.3% in NI [16], 69.6% in England [12] and 77.5% in Scotland [17]. Equivalent male uptake rates were even lower being 67.3% in Northern Ireland [16], 62.4% in England [12], 69.6% in Scotland [17]. Wales public data reported 2022–23 1st dose in 1st year uptake of 12.3% in Wales (all genders) increasing to 69.1% in year 2 of school (72.8% in girls and 65.6% in boys) [18]. This pattern aligns to the sharp declines observed in immunisation rates globally since the COVID-19 pandemic [19].

This decline in HPV vaccination uptake may be due to multiple complex factors [20]. At the beginning of the pandemic, the World Health Organisation (WHO) advised postponement of immunisation programmes to enable IMNs to focus on administration of COVID-19 vac-cines [19]. This has created a backlog of immunisations placing additional workload pressures on IMN teams [21]. IMNs report that provision of education around HPV vaccines in the UK is sub-optimal due to the limited time provided for them to educate students and their guard-ians [18]. While difficult to establish at this early point in time, post-COVID, changes in parental attitudes to vaccination may be influencing HPV vaccination rates. Temsah et al.'s 2021 study [22] conducted in Saudi Arabia found that 45.3% of the population was vaccine hesitant compared to 20% reported in a similar pre-COVID 19 2019 study [23]. Concern about the COVID-19 vaccine may also prompt concern about other vaccines fuelled by unreg-ulated and often misleading social media information [20]. Studies indicate that online nega-tive HPV vaccine video content is more popular than positive video content [24–26]. A Gallup poll from 2022 reported that 38% of people now have no trust in news media, a record high, and are receiving their health information from social media platforms [19]. Given the recent drop in HPV vaccination rates throughout the UK and the issues related to the trustworthiness of information, it is essential to ensure that students and parents are receiving the correct information in order to make decisions regarding HPV.

Studies indicate that the average age of first sexual intercourse in many European countries and the UK is 15–17 years old [27–29] with a significant number of adolescents engaging in unprotected oral sex [30] at a younger age, prior to vaginal intercourse [31]. While adolescent females' knowledge of HPV has improved since the introduction of the HPV vaccine in schools, adolescent knowledge of HPV in general, remains suboptimal [32–34]. A recent UK

study reported that only 25% of university students were aware of the link between oropharyngeal cancer and HPV [33], despite the sharp rise of oropharyngeal cancer worldwide [33, 35], surpassing the rates of cervical cancer in the US [36]. In fact, young people often perceive themselves as having low risk for acquiring STIs compared to other population groups [37].

Northern Ireland has a unique divisive political history and remains a deeply religious society consisting of mainly Catholics (46%) and Protestants (44%) [38–40]. This religious division is reflected throughout Northern Ireland in geographical location, employment and even sporting activities [40]. The majority of schools are religiously segregated into Catholic schools or state schools, attended mainly by Protestants [40]. Less than five percent of the school population attend integrated schools (schools with no religious affiliation) [40]. Despite this division, both Catholic and Protestant churches agree on many aspects related to the morality of sex, stressing abstinence before marriage and instilling a sense of shame and denial, particularly in regard to same-sex relations [41, 42]. Consequently, Northern Ireland's culture is often described as being 'morally conservative toward sexuality, reproduction, and sexual health education'(p19) [43] in comparison to other areas of the UK. Strong religious backgrounds, like those typically seen in Northern Ireland, have been shown to influence condom use, sexual initiation and sexual acceptance [44–46]. Religion has also been shown to be associated with HPV uptake [46–48] with one recent study indicating that highly religious young adults demonstrate lower knowledge of HPV and have lower HPV vaccine uptake compared to less religious young adults [48]. Other factors that have been found to influence HPV vaccine uptake include socioeconomic background and gender [34, 49] with HPV vaccination uptake being generally lower in males compared to females and lower in less affluent populations [50].

Consequently, for this multitude of reasons, providing additional education at a later age (15–17 years old) could provide an opportunity for adolescents to understand their HPV vaccination status and the HPV strains they are protected against. This would also be an opportunity for students who are 16 years old or above and students deemed Gillick competent [14], to self-consent to the vaccine without the need for parental consent. A person under 16 years old can be considered Gillick competent in the UK if they demonstrate a clear understanding of the HPV vaccine and potential consequences of HPV vaccination [51]. In Northern Ireland, if deemed Gillick competent, a young person can legally consent to the HPV vaccine without parental consent [52].

HPV education, to improve knowledge in this age range, has typically been delivered by a professional with a nursing, healthcare or medical background [53]. In some parts of the UK, school nurses have acquired postgraduate qualifications in health promotion/public health and could potentially also be an option for delivery of this HPV education [54, 55]. However, school-based nurses are often part-time or non-existent in schools and there is considerable variety in practice throughout the four countries of the UK [54, 55]. In Northern Ireland, a Christian charity called 'Love for Life' also provides some relationship and sexuality education when requested by post-primary schools [56]. While 'Love for Life' has delivered education in over 75% of post-primary schools, facilitators are not specifically from healthcare backgrounds and therefore focus is more on promoting healthy relationships rather than health education [56].

This study aims to explore 15–16 year old students' perceptions in Northern Ireland regarding whether providing HPV education at this age could influence their intention to be vaccinated and/or future sexual health decisions related to HPV. Students' responses will be captured using Mitchie's Behaviour Change Wheel (BCW) [57], which is a well-established theoretical framework and has been central to the design of numerous interventions related to sexual health including sexual counselling [58], condom use [59] and the use of sexual health services by university students [60]. The BCW framework incorporates the Capability,

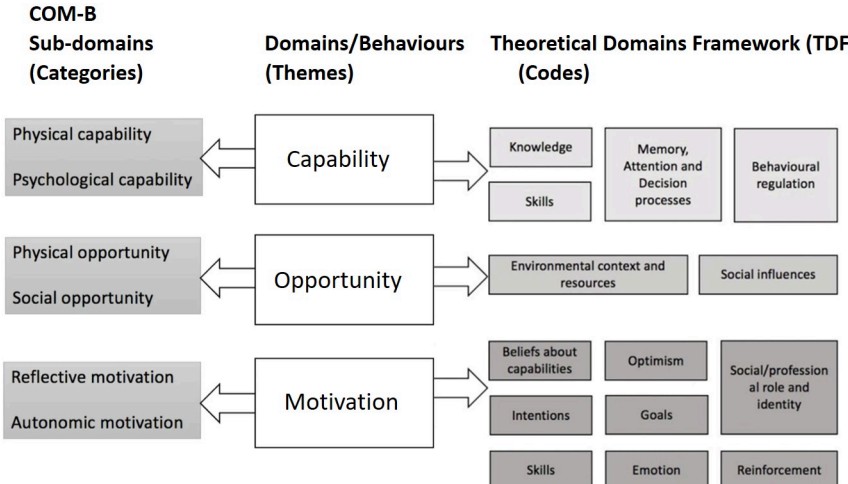

**Fig 1. The COM-B model mapped on to the TDF domains (Note: Autonomic motivation and Automatic motivation are used interchangeably in the context of the COM-B model)** [72].

Opportunity, Motivation, Behaviour (COM-B) model which is a theoretical model for understanding behaviour [57]. The Theoretical Domains Framework (TDF), which can be thought of as an extension of the COM-B model, sub-divides each COM-B category further. Michie et al. [57] describes the 14 TDFs as 'a synthesis of 128 explanatory constructs from 33 theories of behaviour change' (p9). See Fig 1 for cohesion of COM-B model and TDF domains. This model suggests that students' decisions about protecting themselves against HPV is influenced by their perception of their capability, opportunity and motivation. Utilisation of the COM-B model will help to identify potential barriers faced by students and enable the identification of practical measures to facilitate their decision-making regarding protection against acquirement of HPV.

## 2. Method

A qualitative approach utilising focus groups was the method chosen for this study as focus groups have been established as being an excellent way to elicit exploratory information 'concerned with how people make meaning from their experiences in the world' (p924) [61]. They are less time-consuming compared to one-to-one interviews and the interactions between participants in the group can generate an extensive amount of rich data [61, 62]. Studies indicate that teenagers prefer focus groups to individual interviews as they provide a more relaxed, less intimating environment, enabling their peers to be around them to offer additional support [63, 64].

Using the Department of Education's list of 193 post-primary schools [65], stratified random sampling was chosen to ensure representation of schools of different religions, socioeconomic backgrounds and genders as these are all factors which have been found to influence HPV vaccine uptake [34, 47, 49, 66]. An online randomizer [67] was used to select a range of schools representing variation in religion, gender and socioeconomic backgrounds. Subsequently, the selected post-primary schools in Northern Ireland were contacted with the aim of conducting focus groups with diverse groups of students. Through e-mail contact, Head of Schools were asked to pass on details of the study to any teachers that they thought might have students who would be interested in participating in the study. Once the teacher made contact with the

research team, a short information session was held either via Microsoft teams or face-to-face to provide the students with information about the study. Participant information sheets and consent forms were provided to students at this time. Contact information for the researchers was provided to enable teachers, parents or students to ask questions to the researchers prior to consenting to the study. Students were asked to co-sign the consent form with one of their guardians if they consented to be part of the study and return to their teacher one week after the information session. After consent, a Qualtrics link requesting demographic data, was distributed to the students by their teacher to complete prior to participation in the focus group. Students were eligible to participate in the face-to-face focus groups if they were a student in year 12 (15–17 years old) in a post-primary school in Northern Ireland. Consent was confirmed prior to each of the focus groups for participation in and recording of the focus group.

One of the researchers, who held a post-graduate Qualitative Practical Skills Workshop Certificate through Ulster University, facilitated all focus groups while a second postdoctoral researcher in the team, observed and took notes. All four researchers involved in this study identified as female. To optimize the dynamics in each focus group, the researchers aimed to include 4–8 participants per group [68]. Guest et al. (2017) [69] suggest that 4–6 focus groups are likely to yield 90% of all potentially arising themes and therefore the researchers aimed for this minimum number.

The format of the focus group involved the facilitator and students sitting together in a circle, with drinks and light snacks provided for the students. Teachers were not present for any of the focus group discussions. After initial introductions, a brief presentation was displayed on a screen, where the facilitator asked students to consider ten questions about HPV to provide them with context to the topic and enable them to reflect on their pre focus group HPV knowledge. Students were encouraged to provide answers to the questions posed if they felt comfortable doing so. A focus group guide (S1 Appendix) was used to ensure that all focus groups were as consistent as possible. The 'observer' researcher sat away from this main group and did not contribute to any discussion during the focus group but did ask questions of the group at the end if they had any questions. Focus groups were video-recorded with students' consent. The use of video-recording over audio-taping focus group sessions is recommended to ensure that each participants' responses can be identified and to enable extensive analysis of non-verbal cues [70].

## Ethics statement

Ethical approval for this project was granted by the NHS Research Ethics Committee in September 2020 (ID:287358) as a separate part of this study involved focus groups with NHS employed nurses. Individual trust approval was required only for the interview of NHS nurses rather than post-primary school student focus groups. Written informed consent was obtained from the parent/guardian of each participant under 18 years of age.

## 3. Data analysis

Directed content analysis was used to conduct the initial coding and generation of themes. A more structured process compared to conventional content analysis guides this type of analysis [71]. Coding of the data involves using pre-determined codes, categories and themes based on existing models or theories [71]; in this case the COM-B model and TDF domains. NVivo 12 was used to organise the TDF domains into codes, COM-B sub-domains into Categories and the COM-B domains into overarching themes. See Fig 1 for full details.

The focus group guide was produced using the COM-B model and TDF domains to identify the facilitators and barriers that students faced regarding protecting themselves from

acquirement of HPV. Ten of the 14 TDF domains were targeted in the design of the focus group guide. The four TDF domains not targeted were students' personal goals, intentions, behavioural regulation and belief about capabilities regarding HPV to avoid the need for sensitive and personal questions within this focus group environment. Two of the researchers independently organised the data into the pre-determined codes, categories and themes to increase the rigour of the analysis. The two researchers then met and discussed differences in their coding until full agreement was reached.

# 4. Results

Between November 2021 and May 2022, 34 students (24 females; 10 males) from four post-primary schools in Northern Ireland participated in this study within seven focus groups. Each focus group contained between 2–7 students. Twenty-three students (67.6%) indicated that they were Catholic and eight students (23.5%) indicated that they were Protestant. Thirty-two out of thirty-four students (94.1%) described their ethnicity as White Northern Ireland. Seven students (20.6%) did not know if they had received the HPV vaccine. See full demographic details in Table 1.

All emerging themes from the focus groups aligned with the pre-determined categories and sub-domains. See Table 2 for frequency of coding in each TDF domain to highlight the most important TDF domains as recommended by Hsieh & Shannon (2005) [71].

## 4.1 Capability

Psychological capability captures the students' perception of their ability to 'engage in the necessary thought processes, comprehension and reasoning' (p3) [73] in order to protect themselves from HPV and make decisions regarding HPV vaccination.

**Table 1. Demographics of students in each focus group (Note: EMFs = entitled to free school meals).**

| Type of post-primary school | Focus group | Gender | Age | Religion | Ethnicity | HPV vaccination complete |
|---|---|---|---|---|---|---|
| School 1 (Female only Catholic school; >95% Catholic; ~15% EFMs[96]) | Focus Group 1 | Female (n = 6) | 15 (n = 3) 16 (n = 3) | Catholic (n = 6) | White Northern Ireland (n = 6) | Yes (n = 5) No (n = 1) |
| | Focus Group 2 | Female (n = 6) | 15 (n = 3) 16 (n = 3) | Catholic (n = 5) No religion (n = 1) | White Northern Ireland (n = 5) Mixed White and Black African (n = 1) | Yes (n = 5) No (n = 1) |
| | Focus Group 3 | Females (n = 7) | 16 (n = 4) 15 (n = 3) | Catholic (n = 7) | White Northern Ireland (n = 7) | Yes (n = 6) Unsure (n = 1) |
| School 2 (Mixed gender non-Catholic school; >85% Protestant; ~20% EFMs[96]) | Focus Group 4 | Females (n = 3) | 16 (n = 3) | Protestant (n = 2) Christian (not Protestant or Catholic) (n = 1) | White Northern Ireland (n = 3) | Yes (n = 2) Unsure (n = 1) |
| | Focus Group 5 | Males (n = 2) | 16 (n = 2) | Protestant (n = 2) | White Northern Ireland (n = 2) | No (n = 2) |
| School 3 (Mixed gender non-Catholic school; ~70% Protestant; <10% EFMs[96]) | Focus Group 6 | Males (n = 3) Females (n = 2) | 16 (n = 4) 15 (n = 1) | Protestant (n = 4) No religion (n = 1) | White Northern Ireland (n = 5) | Unsure (n = 5) |
| School 4 (Male only Catholic school; >95% Catholic; ~30% EFMs[96]) | Focus Group 7 | Males (n = 5) | 16 (n = 5) | Catholic (n = 5) | White Northern Ireland (n = 4) Indian (n = 1) | No (n = 5) |

**Table 2. Frequency of coding into each TDF domain.**

| | | Coding frequency |
|---|---|---|
| **Capability** | | |
| Psychological capability | Knowledge | 37 |
| Psychological capability | Memory, attention and decision processes | 44 |
| Psychological capability | Psychological skills | 49 |
| **Opportunity** | | |
| Social opportunity | Social influences | 47 |
| Physical opportunity | Physical Environment & Resources | 46 |
| **Motivation** | | |
| Reflective motivation | Social/Professional Role and Identity | 23 |
| Reflective Motivation | Belief about consequences | 13 |
| Reflective Motivation | Optimism | 6 |
| Automatic Motivation | Emotion | 26 |
| Automatic Motivation | Reinforcement | 0 |

**4.1.1 Knowledge.** All students indicated that their knowledge was insufficient regarding HPV and that this topic was not discussed in detail at school. Most students were not aware that HPV was an STI, though those who were currently taking biology were aware of HPV being a virus. The majority of students expressed that their education regarding STIs in general was inadequate. Some students in each focus group demonstrated awareness of the link between HPV and cervical cancer and had an awareness of cervical screening. However, none of the students were aware of the other types of cancers associated with HPV. Students demonstrated a particular lack of knowledge regarding the types of HPV-associated cancers that males could acquire, thinking of it as a 'female issue' rather than affecting all genders. The majority of students were unaware that the HPV vaccination was now being offered to boys.

Female students being educated in a female only Catholic school felt that some students are very misinformed about the HPV vaccine. Reflecting on when they received their vaccination, students from all schools indicated that they could not recall any meaningful discussions within the school with either teachers or with the IMNs prior to the HPV vaccination. They indicated that they received so many vaccinations that they were confused regarding which vaccines they had received. While male students in the focus groups were not offered the HPV vaccine, some were unaware of whether or not they had received it in school.

Students felt that by participating in the focus groups, they had acquired a lot of knowledge and felt more informed about HPV vaccination and transmission routes. All students were highly in favour of more HPV education. Students in all groups felt that, as 15–16 year olds, there was no information that should be off limits in HPV education and that they needed to know everything about HPV and other STIs. They felt that it was important that all forms of transmission be discussed openly including risks associated with oral, anal and vaginal sex. They wanted to understand preventative measures, symptoms of HPV and management of HPV or another STI.

Many female students (of both religions) indicated that not all students would understand medical terms like oral sex or anal sex. They felt that it was important that the facilitator used non-scientific terms that students might know alongside the correct scientific terminology. However, male students had mixed opinions regarding whether non-scientific terms should be used in this educational setting but agreed that by only using scientific terms, the conversation would also be more formalised. Males felt that using non-scientific terms would be unprofessional and reduce the importance of the topic.

**Table 3. Students' views on their current knowledge of HPV and what content should be included in HPV education for their age group.**

| |
|---|
| *'I had no clue until I sat down today that HPV was an STI and not just cervical cancer. I thought that it was something that maybe you were unlucky enough to develop, not catch.' (S10, Focus Group 2; Female only Catholic school)* |
| *'Even when we got the letter about this, I was like, which one is the HPV? Because we get so many. . . it can be really confusing.' (S2, Focus group 1; Female only Catholic school)* |
| *'I went to get it [HPV vaccination], one of my friends was like, I'm not getting it. And I was like, why? And they were like, because you might get pregnant from it.' (S15, Focus group 3; Female only Catholic school)* |
| *'I'm not entirely sure, but does it affect males as well?. . . I think there's a lack of education around males affected.' (S13, Focus group 3; Female only Catholic school)* |
| *'If we were actually taught all the facts, I feel like most people actually would choose to get it [HPV vaccination]'. (S15, Focus group 3; Female only Catholic school)* |
| *'You should be open about it [HPV transmission routes]. People our age sort of know. It would be different in the younger year group. But our year, you know more about. . . and it wouldn't be as sensitive to talk about that with us.' (S23, Focus group 5; Males from Mixed gender non-Catholic school)* |
| *'It can be awkward at times. But. . .if you are all mature enough. . . like we are all sitting here having this conversation and no one is sitting here giggling or getting on because someone mentions sex.' (S29, Group 6; Mixed genders from Mixed gender non-Catholic school)* |
| *'It would be more relatable [to use non-scientific sexual terminology] but people need to understand. . . if you talk like that it will just remove all stages of importance.' (S34, Focus group 7; Male only Catholic school)* |
| *'It would just be unprofessional [to use non-scientific sexual terminology].' (S30, Focus group 7; Male only Catholic school)* |
| *'I think just like what would happen if you get HPV and ways to protect against it, would be good to be included in the education.' (S4, Focus group 1; Female only Catholic school)* |
| *'Some people maybe don't actually understand about condoms.' (S13, Focus group 3; Female only Catholic school)* |
| *'All your other types of STIs, no one gets taught about any of that, or what happens when you do get those, or what to do.' (S18, Focus group 3; Female only Catholic school)* |
| *'And how you can get it. Because that was something I didn't know about.' (S5, Focus group 1; Female only Catholic school)* |
| *'I remember thinking it was because I had a Spanish oral and I thought, that must be like. . . [points at mouth].' (S19, Focus group 3 (Female only Catholic school) recalls a funny incidence where the term 'oral sex' was used in class and how she worked out what it meant.)* |

All students felt that there would need to be restrictions on the detail provided to younger students and topics would need to be adapted to be age appropriate. All students expressed that revisiting and building on information regarding HPV and other STIs, was important to do each year from age 12 to 16 years old and would help to normalise sexual health topics, reducing associated stigma. See Table 3 for students' comments regarding this aspect.

**4.1.2 Memory, attention and decision processes.** All students felt that a short face-to-face education format was essential to deliver this HPV education to aid attention and memory. The majority of students indicated that a small group size of 6–10 would be ideal when receiving this education to optimise discussions with the facilitator. They felt that it should be organised so that students are with some friends or others that they know well. They felt that this environment would encourage a more open debate if there was disagreement within the group. However, they recognised that this number may not be practical to implement and therefore agreed that a class size (15–30 students) would also work well; especially their form class. They felt that there is trust already established within a form class setting, and that this would be a safe alternative environment. All students agreed that larger class sizes in assembly halls would not be appropriate.

Some students in a female only Catholic school expressed that they would not feel comfortable receiving this education with boys present, indicating that it would restrict their ability to ask questions. However, the majority of students in this female only Catholic school felt that

**Table 4. Students' comments on aspects of the group environment for HPV education.**

| |
|---|
| *'I think boys and girls should be separated for that kind of talk, just because you'd be less comfortable maybe talking about it.' (S6, Focus group 1; Female only Catholic school)* |
| *'Even in primary school when we were given the talk about puberty, the boys were put in one room and got it, and the girls were put in the other room and got it. So there wasn't any carry on!' (S5, Focus Group 1; Female only Catholic school)* |
| *'Separation is such an old thing . . .it creates a stigma. . ..if you split them it makes it a big deal. . .at least if they're doing it together, they can see what each other think about what they are saying. ' (S10, Focus Group 2; Female only Catholic school)* |
| *'It would also be better if I am with someone I know, so that I'm not in a room full of strangers with a possibly uncomfortable topic.' (S34, Focus group 7; Male only Catholic school)* |
| *'You're not going to talk if it's a whole year group. . .you can't have conversations. It's more like, hand up, say your point and that's it. . .and they are like, do you have any questions? No one asks. . .there's a hundred plus eyes on you! (S16, Focus group 3; Female only Catholic school)* |

splitting up genders was outdated and that students of all genders should be taught together. Ultimately, they felt that it was important that genders were not divided into groups as they needed to learn together. They agreed that the best option would be to have a mixed gender lesson but then have a short debrief session afterwards with separate gender groups.

Students from mixed gender schools felt that having the groups mixed was important and that they would not feel awkward with mixed genders. Student comments are summarised in Table 4.

All students liked the idea of a more informal approach like having a 'chat' rather than someone delivering powerpoint slides. The majority of students commented on the importance of interactive activities like quizzes, practical condom demonstrations, snappy statistics and groupwork activities to aid their learning. Students recalled external facilitators who did practical activities and indicated that they remembered most from these enjoyable interactive classes. The majority of students indicated that they would not read a leaflet if provided.

Female students discussed the use of personal stories through watching videos by young people affected by HPV as being very impacting and influencing and felt that incorporating this into the education would be effective. However, male students felt that, if using real life stories, facilitators need to take care not to make the details too scary or intimidating though they agreed that hearing about a young person's experience might be helpful. Some male students also indicated that boys tend to ridicule the people talking on videos.

All students agreed that it would be important to have the opportunity to ask questions anonymously related to the taught educational content. Students described technology, like Slido polls, which are available to assist anonymity when asking questions during the face-to-face education. A phone chat option with the facilitator post-education was discussed but students were definitely not keen on using the phone but would consider texting a facilitator.

The majority of students liked the idea of following up the face-to-face HPV education with some form of dedicated 'snappy' social media area embedded in Instagram or Tiktok. Students indicated that they rarely click on links provided during class and therefore they would not recommend their use. Students talked about the NHS designing an app to supplement this education similar to the COVID-19 app. However, some students felt that most students would not bother with social media after the class was complete. Posting supplementary information on Google classroom was not perceived favourably as students felt that they 'missed everything' on that site. Some commented that not all students would want to see graphic images of tumours but agreed that components like this could be embedded into Instagram for students to choose to view if they wanted. See Table 5 for students' comments regarding this aspect.

**Table 5. Students' comments on their preferred format for the HPV education.**

| |
| --- |
| *'I think it would be good if it was a conversation rather than a presentation.' (S14, Focus group 3; Female only Catholic school)* |
| *'. . .the facts and figures. I found the figures were so. . . because it's just there and they are so in your face and it was like, you have to take it on board.' (S2, Focus group 1; Female only Catholic school)* |
| *'We watched a film in RE. . .there was this guy, and he was talking to someone. . .as soon as he appeared on the screen in his suit, everyone just started laughing and making fun of him, because he was sat there on the screen. And I don't know what it was, maybe the suit was too big, but everyone was making fun of him.' (S32, Focus group 7; Male only Catholic school)* |
| *'I think it should be short and concise because if we get too much information all the time we'll just ignore it.' (S30, Focus group 7; Male only Catholic school)* |
| *'I think the demonstrations of how to put a condom on and how it keeps you safe is probably good for people in our year group, just to keep them safe.' (S33, Focus group 7; Male only Catholic school)* |
| *'People our age are embarrassed very easily talking about stuff like this. And they would also feel embarrassed to talk on the phone. Whereas if they are texting they wouldn't feel that embarrassed.' (S23, Focus group 5; Males from Mixed Gender non-Catholic school)* |
| *'You go to school and you learn about these things and then you just forget it when you go home. But when you see it on social media it reminds you, and you're like, oh yeah!' (Focus group 1; Female only Catholic school)* |
| *'If you come across it on Instagram you're more likely to read it, than if you see a link and go, oh I'll click that and read it.' (S8. Focus group 2; Female only Catholic school)* |

**4.1.3 Psychological skills (Cognitive & Interpersonal).** Students in all groups discussed the difficulty in trying to educate 12-year-old students about HPV at the time of vaccination due to their lack of maturity and inability to comprehend sexual relationships. All students felt that younger students tend to giggle and laugh during these types of classes. Students also felt that hearing about cancer at that age, would be quite scary.

All students explained how the form provided at the time of vaccination was not a suitable format to use for children aged 12–13 years old. They suggested that alternative formats like a short presentation or posters, would be more appropriate for students at this time. While students in all groups agreed that the information would not be as relevant to them at that time, they all strongly expressed that they would have liked to hear the basic facts about what the vaccination was protecting them against when they received it.

Students expressed mixed opinions regarding the age where more comprehensive information should be provided to students e.g. specific transmission routes and sexual behaviours associated with transmission. Around half of the students agreed that year 12 (15–16 years old), was the right time to receive more comprehensive information about the various aspects of HPV. They were concerned that providing comprehensive detail to 14-year-old students would be too scary for them. One female student (from a non-Catholic mixed gender school) even indicated that they felt that providing information at this age would encourage students to become sexually active.

However, the other half of the students felt that providing comprehensive education to students at 14 years (year 11) would be a more suitable option. They indicated that while not many students were sexually active at 14 years old, some students were and they needed to be safeguarded. They felt that timing comprehensive education at 14 years old would ensure that when they became sexually active, they would have the information to protect themselves. They felt that the content could be the same as year 12, but the language adapted slightly for that age group. At this age, they felt that students would understand the importance of this information and want to engage in conversations about this topic.

Gender or religious denomination did not appear to be a factor influencing individual students' opinions regarding this aspect of the HPV education. See Table 6 for students' comments regarding this aspect.

**Table 6. Students' comments on the differences in psychological capability based on student age.**

| |
|---|
| *'We are mature. If someone said sex in the room we wouldn't be laughing. But if someone was first year, second year. . .they mightn't understand it and they might just think it's funny instead of just taking it seriously and something they may need to listen to.' (S30, Focus group 7; Male only Catholic school)* |
| *'You can't really tell a twelve year old all of these morbid facts.' (S34, Focus group 7; Male only Catholic school)* |
| *'The information that's given is not user friendly. . .you can't hand that to a young person and expect them to read through that. . .maybe our parents can read through that but I think it would need to be edited down for us to be able to understand that.' (S13, Focus group 3; Female only Catholic school)* |
| *'Talk to us about it. We all sort of went in, got a vaccine that we didn't really know much about.' (S22, Focus group 4; Females from Mixed gender non-Catholic school)* |
| *'Maybe a wee bit more light-hearted when they are twelve.' (S22, Focus group 4; Females from Mixed gender non-Catholic school)* |
| *'I think maybe fourteen because . . .by this age it might be too late for some people if they don't know about the risks.' (S23, Group 5; Male only Catholic school)* |

All students agreed that at 16 years of age, they were mature enough to make their own decisions about receiving the HPV vaccination and should be able to self-consent if offered the vaccine in year 12. They felt that having to ask their parents at this age would compromise their privacy. Female students felt that 15 year olds were mature enough to make their own decisions though male students felt that some 15 year old students may not have the capacity to self-consent. While many students were aware that legislation may not enable 15 years olds to self-consent, they also discussed the lack of education provided to them regarding their legal rights.

All students agreed that at 15 years old, it would be important for students to be assessed for their understanding of the information through chatting to the IMN prior to self-consenting to a vaccine. The majority of students felt that 14 year old students would not have the capacity to self-consent and make their own decisions due to lack of maturity.

Some students (from both genders and from both religious backgrounds) indicated that they felt that it was important that their parents knew that they were being offered the education and opportunity to self-consent to the HPV vaccination. They felt that it was important to include their parents and discuss it with them even if the decision was ultimately up to them.

Students stressed that it would be important for there to be a gap between delivering the education and taking up the HPV vaccine to provide students with time to reflect and make their decision after the education. See Table 7 for students' comments on this aspect.

## 4.2 Opportunity

This section captures students' perspectives regarding all of the external factors which they feel have impacted their access to HPV education and/or HPV vaccination. Michie's COM-B model describes opportunity as being related to social influence (culture) and/or physical environment [73].

**4.2.1 Social influences.** Students from all focus groups discussed how their parent/guardian were the primary decision-makers regarding all of the vaccinations that they had received to date including the HPV vaccination. The role of their parents in this decision-making process was one of high importance when they were younger. However, students indicated that this resulted in them taking a very passive role in their vaccination choices in the past and admitted that they had not really even considered what the vaccinations were protecting them against.

Most students felt that their parents were very supportive of them learning about HPV but many students felt that parents of other adolescents may not be so supportive. Students added

**Table 7. Students' perceptions regarding consent for the HPV vaccination.**

| |
| --- |
| *'I think at fifteen, sixteen you are old enough. . . your parents shouldn't have a say. . .maybe when you were younger and not exposed to this yet. But I think you are old enough and mature enough to know all this.' (S6, Focus Group 1; Female only Catholic school)* |
| *'It's for your own wellbeing and health. I think it [consent] should ultimately be down to us.' (S1, Focus group 1; Female only Catholic school)* |
| *'I think if you've been given the right education on it, then you should be allowed to consent yourself.' (S29, Group 6; Mixed genders from Mixed gender non-Catholic school)* |
| *'It allows them [parents] to know what's going on in school, for them to be involved a bit. Because you could just not tell them and then go on and do it. . .I think it's good to. . . maybe not give it to them to get consent, but to be involved with it and to know what you're being taught in school.' (S2, Focus group 1; Female only Catholic school)* |
| *'Schools can be held liable for that sort of thing as well, I guess. Because if something happens to a fifteen year old when they get that, it can always be brought back to the school, I guess. I don't know.' (S13, Group 3; Female only Catholic school)* |
| *'It's also the fact that I didn't know until today that you could go at sixteen to get a vaccine. I got my COVID vaccine, but I thought that was just a special case. But there is no education on what you can actually do when you are sixteen, what freedoms you have.' (S30, Focus group 7; Male only Catholic school)* |

that they did not feel that their parents would necessarily be supportive of them being sexually active. Reasons discussed through the groups for potential lack of parental support included concerns regarding the safety of vaccinations and a perception that HPV education would promote promiscuity.

Students in all groups felt that parents may not always have the education needed to make informed decisions for their children. They stressed the importance of educating parents to ensure that they can make informed decisions for their children. They felt that parents considered HPV a female issue and therefore may not understand the importance of HPV vaccination for males.

Some students talked about their parents often making decisions based on their own life experiences. Influencing factors included having a family member with a HPV-related cancer, their parent's age and their parent's own experiences when they were at school.

Some students highlighted the challenges associated with teaching parents as there would be no option to mandate education for them. They felt that educating students now would ensure that parents are educated in the future. Most students felt that their parents found it difficult to talk to them about sexual health.

Of note, students from the male only Catholic school commented that if parental consent had not been required for the focus groups, a much greater number of students would have participated. See Table 8 for students' comments regarding this aspect.

All students felt that HPV and other STIs are stigmatised both in the Northern Ireland media and the school setting and that this impacts their opportunities for STI education.

Both male and female students felt that their sexual health education was negatively impacted by being educated in a Catholic school. They also felt that parents and schools with a strong religious affiliation, particularly Catholic schools, would have objections to HPV education being taught in schools.

Students from the male only Catholic school felt that their parents would be 'furious' if detailed sexual health education was provided to them before they were 16 years old. Students attending Catholic schools felt that there was an expectation that they would abstain from sexual interactions and therefore would not need to know about aspects of HPV as a result. Religion was not raised as an issue by students attending non-Catholic schools. See students' comments in Table 9.

**Table 8. Students' comments on the influence of their parents regarding their opportunity for HPV information and HPV vaccinations.**

*My mum was like, you're getting it. It will protect you. . .so I am happy she made me get it now. She said it will protect you against cervical cancer. You have to get it. So I was like, right, OK, fair enough, I will!' (S8, Focus group 2; Female only Catholic school)*

*'. . .they [parents] might not know that information, because obviously the school was very restricted when they were at school. . .so they wouldn't be able to teach you anything.' (S1, Focus group 1; Female only Catholic school)*

*'We are basing our decisions off the information that the school is giving us, but they are basing it off just their life. They haven't been sat down and told. So we're going to have different opinions.' (S3, Focus group 1; Female only Catholic school)*

*'I didn't get it the first year it came out, because my mum didn't have the information. And she was like, I'm not really sure. And she'd heard all these stories from friends. . . the school continually offered it, that's why I got it the second time round, because the school offered it again.' (S13, Focus group 3; Female only Catholic school)*

*'I said that to my dad this morning and he was like, is that not for AIDS and all? They didn't know.' (S19, Focus group 3; Female only Catholic school)*

*'My aunt had cervical cancer the year that I got that. . . when that form came out, they were like, yeah, you should probably get that.' (S10, Focus group 2; Female only Catholic school)*

*'It would depend on how they [parents] have been brought up as well. That would all factor into what they think about it.' (S7, Focus group 2; Female only Catholic school)*

*'I feel like they expect you to know. . . like learn about it in school.' (S18, Focus group 3; Female only Catholic school)*

*'It's something even our parents don't want to really talk to us about. No one really wants to say.' (S22, Focus group 4)*

*'It was the fact that we had to go home and get consent forms from our parents that there isn't triple or quadruple the amount of people right now [in the focus group].' (S34, Focus group 7; Male only Catholic school)*

*'. . . it's always been girls that have got it, so why do the guys need to get it? There's also that stigma that might be hard to break.' (S30, Focus group 7; Male only Catholic school)*

**4.2.2 Physical environment and resources.** All students felt that the school environment provided the best opportunity to receive all of their HPV education as it was convenient to all adolescents. All students indicated that HPV education and sexual health education is not prioritised throughout their existing school curriculum. They felt that HPV education should be an integrated component of a larger sexual health education programme and be aligned to teaching about other STIs. All students felt that HPV education should be a mandated component of the normal school curriculum. Students talked about the importance of a consistent

**Table 9. Students' perceptions of the influence of social stigma and religion on opportunities for HPV education and HPV vaccination.**

*'It [sex] is not normalised here. It's secretive and frowned upon and at our age it shouldn't be.' (S16, Focus group 3; Female only Catholic school)*

*'It's like a wall is built up around it [sex] and why would you talk about it? . . .they [adults] are embarrassed' (S21, Focus group 4; Females from Mixed gender non-Catholic school)*

*'I don't know how it would be in say like a Protestant school, but I think there is a bit of a taboo around the whole sex topic in Catholic schools. Even in other Catholic schools in the area.' (S5, Focus group 1; Female only Catholic school)*

*'The religious element. . .the fact that we shouldn't be getting the vaccine for something that we shouldn't be putting ourselves in the situation to get in the first place. Because. . . I think that's the reason we don't have sex education here either. Why would we need it? We are all girls, we're Catholic. . .' (S10, Focus group 2; Female only Catholic school)*

*'Teachers are maybe scared to talk about it because of the repercussions in the school. So it's not really individual teachers' fault. It's the fault of higher up. . . because it is a Catholic school and teachers are maybe scared to give you that information, and the correct information. (S13, Focus group 3; Female only Catholic school)*

*'Personally, I think that if you are going to have a talk in schools, you should teach. . .when they are fifteen. But I don't think that's really realistic, especially for Catholic schools. It would probably be a lot easier to include it in unit two instead, even if it would be less effective, because in unit two you're over sixteen so you wouldn't have as many parents being furious about it.' (S32, Focus group 7; Male only Catholic school)*

**Table 10. Students' beliefs on the importance of consistent HPV education in all post-primary schools.**

*'It should be part of a larger thing. . .there's a massive lack in sex education in schools all across Northern Ireland. If, for example, integrated schools in Northern Ireland were receiving that education, then Catholic schools should be receiving that as well. Because it's no different.' (S13 Focus group 3; Female only Catholic school)*

*'Yeah, I think if you don't really give an option. I think everybody is old enough now. It just becomes part of normal education.' (S1, Focus group 1; Female only Catholic school)*

*'Different teachers would have different opinions. . .you can see that in other subjects. You can see different opinions coming out. But this is serious and you can't have people's opinions affecting your decision on this.' (S13, Focus group 3; Female only Catholic school)*

*'We had Love for Life. It was sort of. . . it was about everything but it was more in a twelve year old friendly way.' (S22, Focus group 4; Females from Mixed gender non-Catholics school).*

*'All your other types of STIs, no one gets taught about any of that, or what happens when you do get those, or what to do.' (S18, Focus group 3; Female only Catholic school)*

*'PD is the most stupid thing ever. . .it's about mental health but they're going about it the complete wrong way. . .like when you're doing work, take a ten minute break and you'll feel fine. . .there is time to teach us about this type of topic in PD but they don't. . .but this should be a topic.' (S16, Focus group 3; Female only Catholic school)*

*'In careers we are doing nothing. And in that time where we are literally just goofing about, we could be learning about things like this and benefitting from it, easily.' (S34, Focus group 7; Male only Catholic school)*

approach within this mandated education in all post-primary schools in Northern Ireland regardless of religious affiliation or school ethos. One student, from an ethic minority, highlighted the importance of equality in sexual health education as they felt that some ethnicities do not openly talk about any sexual health related issues.

Students explained how HPV education and other sexual health topics could fit well into the Professional Development (PD) classes or replace some Careers classes throughout the curriculum. Male students felt very strongly that they should not lose 'games' time to attend these classes but that they should be built into the curriculum.

Some students had received relationship education through 'Love for Life', an external specialist provider. They described how these classes focused on relationships but also touched on STIs. They explained how 'Love for Life' built on these topics and increased their complexity each year to align with their development. However, they all agreed that the detail covered regarding STIs in these classes was insufficient. Some students suggested that 'Love for Life' could adapt their sessions to incorporate this type of detail. Students attending the female only Catholic school did not receive this education. See student's comments in Table 10.

Students felt that, if students were re-educated in year 12 (age 15–16 years), then another opportunity would need to be provided within the school for HPV vaccination shortly after the education. They indicated that practical aspects would negatively affect students' opportunity to get the vaccination outside of the school environment. This included inability to travel to their GP, lack of initiative to book an appointment with their GP and possibly having to converse with their parents regarding why they want the HPV vaccination. See students' comments in Table 11.

## 4.3 Motivation

This section captures students' perspectives regarding their motivation, which are the internal processes, which influence their decision making and behaviours [65]. Michie et al. (2011) [74] describe two types of motivation; reflective motivation and automatic motivation. Reflective motivation involves motivation due to reflection on past events while automatic motivation involves 'our desires, impulses and inhibitions' (p3) [65].

**Table 11. Students' views on the importance of the school as a physical space for HPV education and HPV vaccination.**

| |
| --- |
| 'I think that would be worse because it's like, oh I have to book it, go to the doctor's. I'd be like, just leave it. It's fine.' (S2, Focus group 1; Female only Catholic school) |
| 'No that would probably. . . students couldn't be bothered. . .I was saying I was going to get my Covid vaccine for ages and then I just never got round to it. But when it came to school it was easier to get it that way.' (S16, Focus group 3; Female only Catholic school) |
| 'It kind of needs to be in school because no one . . . genuinely no one, however good a teacher you are, you cannot convince a sixteen year old to phone up his GP, get a lift, spend all the time there, do the forms, there's no way you can convince them.' (S32, Focus group 7; Male only Catholic school) |
| 'It would be a lot more hassle that way. So less people would be less likely to go to that effort to do it. . .and then there would be the questions of why you are going to that effort. (S9, Focus group 2; Female only Catholic school) |
| 'It's out of the way. And getting into your GP is already hard enough.' (S30, Focus group 7; Male only Catholic school) |
| 'And sixteen year olds don't want to tell their parents.' (S19, Focus group 3; Female only Catholic school) |

**4.3.1 Reflective motivation.** *4.3.1.1 Social role and Identity.* All students felt that information about HPV and other STIs was very relevant to their lives and important to understand. They indicated that students generally were becoming sexually active at 15–16 years old, with a smaller number of students being sexually active at 14 years old.

Female students talked about how they were motivated by stories of celebrities who had HPV associated cancer. They talked about watching Jade Goody's life with cervical cancer and felt that these types of TV programmes helped them to realise that this could happen to them, increasing their motivation to learn about HPV. Many students expressed hearing about HPV through social media platforms but were cautious regarding the large volume of misinformation on social media. Many of these students expressed a lack of trust in social media information related to this topic. They also highlighted negative media attention regarding cervical screening. Previous experiences of HPV-related cancer in their families also increased students' motivation to want to receive HPV education and the HPV vaccinations. See Table 12 for students' comments regarding this aspect.

*4.3.1.2 Belief about consequences.* All students felt that HPV education would help them to understand the consequences of acquiring HPV as, prior to the focus group, they were largely unaware of most of the consequences. Without any additional education, female students

**Table 12. Students' comments on the social influences that motivate them to want school-based HPV education.**

| |
| --- |
| 'You should be receiving information on nearly every single STD that you can get, because around this age you are probably sexually active or starting to be.' (S24, Focus group 5, Males from Mixed Gender non-Catholic school) |
| '. . .it was so sad. . .I remember watching it and. . .she [Jade] was advocating for this to come out. . .because she was like, it completely destroyed her life and her son's life. . . it made you realise that it can happen to anyone.' (S2, Focus group 1; Female only Catholic school) |
| 'It makes it more realistic. When somebody gives a talk you might think, that might not happen to me. But when you hear a personal story and how it impacts somebody in their life, it makes you think about it more and be more aware of it.' (S5, Focus group 1; Female only Catholic school) |
| 'It's such a big thing now, as well, because there's so many people you hear of getting it, as well. . .on social media you see there's more awareness around it now than there ever was.' (S22, Focus group 4; Females from Mixed Gender non-Catholic school) |
| 'I think there's a lot of rumours on social media as well. . .you never really know what's accurate.' (S17, Focus group 3; Female only Catholic school) |
| 'Part of it is the culture where you just naturally agree with the video that's being put up. And then obviously it happens on every platform, on Twitter and things like that. When I go now and look, there's a lot of misinformation on there with people, as well.' (S13, Focus group 3; Female only Catholic school) |

**Table 13. Students' perceived consequences of being educated about HPV.**

| |
| --- |
| *'I think if you are starting to educate people like us, when we are parents, we are OK with it, and so maybe further down the line with this education, it might be normalised.' (S30, Focus group 7; Male only Catholic school)* |
| *'Without further education people would think that they were safe because they've got the vaccine.' (S6, Focus group 1; Female only Catholic school)* |
| *'I think the education. . .it would make me more likely to actually go get it, because I would understand more about the seriousness of it. . .so I think it would actually make me a lot more likely to go get it, rather than be, oh I have the vaccine, I don't have to get it, I'm fine.' (S11, Focus group 2; Female only Catholic school)* |
| *'I think it would make us all more aware, for growing up and maybe becoming sexually active, you have that in the back of your mind that you need to be careful and not just careless about it.' (S5, Focus group 1; Female only Catholic school)* |
| *'I think you are better to be educated on it and then you are going out into the world knowing things that you wouldn't have been taught previously. You'd be a bit more confident in yourself.' (S2, Focus group 1; Female only Catholic school)* |

demonstrated a reasonable awareness of the importance of cervical screening in preventing cervical cancer. However, until these focus groups, students were largely unaware that males could also develop HPV associated cancers. Students indicated that even information gained from these focus group sessions increased their motivation to want to be vaccinated and protect themselves from possibly developing HPV-associated cancers.

Some students commented that educating them would ensure that they were able to make their own informed decisions about vaccinating their own children in the future. This was perceived as an important and positive consequence of this HPV education.

All students expressed that having more knowledge through additional HPV education, would encourage safer sexual behaviour. All students felt that further HPV education would positively influence their attendance at any future HPV screening being offered to them. Some female students (from the female only Catholic school) felt that, with student's current level of education, many students would not attend screening when offered it in the future. They felt that some students who were vaccinated might feel that they were fully protected against HPV and not understand the importance of attending for screening.

Male students indicated that they would take the HPV vaccine if it was offered to them in school today. See student's comments regarding this aspect in Table 13.

*4.3.1.3 Optimism.* The majority of students felt that barriers may arise when trying to implement HPV education into schools. The most frequently cited barrier was a lack of support from some parents, particularly those with strong religious backgrounds. Many students felt that schools would be resistant to mandating HPV education into the curriculum unless supported by legislation from the Education Authority.

However, students indicated that despite the barriers discussed, they could all be overcome to enable the introduction of this HPV education. See Table 14 for students' comments regarding this aspect.

**4.3.2 Automatic motivation.** *4.3.2.1 Emotion.* The majority of students indicated that talking about STIs like HPV can be embarrassing for them and for the facilitator of the education. Consequently, the majority of students felt that the facilitator delivering the education should be an expert in this field who is external to the school. They felt that having open discussions would be easier with a person that they did not know but who had a strong healthcare background. All students indicated that the immunisation nurses are aptly placed to provide this education and rated their expertise as being of high importance. They felt that students pay more attention to and are more motivated by an external visitor and take the class more seriously. Students talked about how embarrassed most teachers would be talking about this

**Table 14. Student optimism regarding the implementation of HPV education.**

*'It could be quite hard to get it past the Education Authority and then to get schools to approve. And then parents. Because some parents maybe wouldn't agree and some students might not want it. And it would be quite hard to get all of that. It would take a long time, I would say.' (S2, Focus group 1; Female only Catholic school)*

*'Yeah, I think there always will be barriers. Parents. School. As we were just talking about, Catholic school, different schools. . . parents will always want to have a say in what their child is being taught. So that is a big barrier. And maybe the child's perception and the parent's perception on what you should be taught is going to be very different. And that does get complicated.' (S13, Focus group 3; Female only Catholic school)*

*'If they are just teaching about what the disease actually does, then no I don't think there would be much resistance. But if you teach about how it's transmitted and what you can do to prevent transmission, some religious or some conservative, I guess, families might have some issues with their child being exposed to that.' (S32, Focus group 7; Male only Catholic school)*

subject and felt that teachers would not have expertise in this area. Most students indicated that they would not feel comfortable asking questions to teachers.

Students discussed how the 'Love for Life' group could alternatively incorporate this aspect into their teaching as they appeared to have very good knowledge and were energetic and enthusiastic.

Some students indicated that while IMNs would be appropriate to delivery this education, so would some of their science teachers. Males from various schools indicated that the facilitator would need to be able to assert themselves to prevent disruptive behaviour. They felt that the facilitator would need to have some experience in dealing with teenagers in this environment to control the class.

While three of the four schools had school nurses, only students from one school felt that, with the right training, the school nurse would be an appropriate person to deliver this education. However, most students felt that school nurses did not possess the skills or knowledge.

Most students felt that 25–30 years old would be the perfect age for a facilitator, as they would have a greater understanding of the student's social culture. However, all students felt that the age of the facilitator was not as important as other characteristics; students valued expertise, openness, passion and being non-judgemental as being the most important attributes of the external facilitator. Some students liked the idea of young health professional students teaching this material. See Table 15 for students' comments regarding this aspect.

Female students from the female only Catholic school indicated that fear was a strong motivator driving their desire to be vaccinated and educated about HPV. Individual experiences had contributed to that fear including media coverage regarding errors in cervical screening in Ireland and also negative experiences of family members contracting HPV and developing cancer.

Some male students felt that some amount of fear needed to be generated by the HPV education in order to encourage all students to pay attention to the education. One student indicated that it should be 'just below scaring for it to work' (S30).

Some female students from the female only Catholic school, mentioned fear of needles and cervical screening as being possible negatively influencing factors in some students' decisions regarding the HPV vaccination. See Table 16 for students' comments regarding this aspect.

## 5. Discussion

This qualitative focus group study, informed by the COM-B model, explored students' perceptions of the facilitators and barriers associated with protecting themselves from HPV acquirement.

**Table 15. Students' comments on the characteristics of the ideal facilitator for HPV education to minimise embarrassment and increase motivation.**

'We do so many things in the form class that you don't take it all in. But when it's a special visitor, you pay more attention, I think.' (S9, Focus group 2; Female only Catholic school)

'They [IMNs] are probably passionate about it and that would come across as they were speaking about it. And you'd want to listen more and you'd want to learn more about it. . .if it was someone who came in [from outside the school], it might stick a lot more. We might remember it in the future.' (S11, Focus group 2; Female only Catholic school)

'If we are going to go down the road of education in schools, I would say set a day or something, or actually bringing in the vaccination nurse to talk about it. Because if your biology teacher is starting in the middle of the class, it's mundane, it's normal. Whereas if you had something like a whole day taken up with experienced people, it would get them to realise and focus.' (S30, Focus group 7; Male only Catholic school)

'That's why I think Love for Life is a good thing, so it's not one-on-one, it's a group of people. And it's not awkward. You don't know the people personally.' (S26, Focus group 6; Mixed genders from Mixed gender non-Catholic school)

'There's certain things you learn through experience of being a teacher, and one of them is how to control your class. . .putting this subject in the hands of someone who has no idea how to control a classroom, it wouldn't work.' (S32, Focus group 7; Male only Catholic school)

'It is much easier whenever they [the facilitator] are younger. It feels more with the times.' (S28, Focus group 6; Mixed genders from Mixed gender non-Catholic school)

'If it comes from a nurse, you would hope that it would come from a factual perspective. I think there's more of a trustworthy sense there, than if it's coming from a teacher.' (S13, Focus group 3; Female only Catholic school)

'I have never seen her [school nurse] do anything past a paracetamol. It's, here's a paracetamol or a wet towel, and if that doesn't work you go home.' (S32, Focus group 7; Male only Catholic school)

## 5.1 Capability

Students indicated that they do not currently have adequate knowledge about any aspects of HPV to inform decision-making regarding acquirement of HPV, consistent with previous studies [75]. While students indicated that they had some awareness of HPV-associated cervical cancer, students perceived HPV as a female issue and were unaware of the male cancers associated with acquiring HPV. These findings are consistent with a recent study by Franca et al. (2022) [76], who surveyed university students in the US. From 862 students (570 female; 292 male), 70% were aware that HPV can cause cervical cancer, compared to only 34–39% being aware of HPV as a causative agent for oropharyngeal, penile or anal cancer. Another study carried out in the US [35] in a military base also reported higher awareness of HPV as a causative agent for cervical cancer but poor awareness of the link between HPV and oropharyngeal cancer. Given that 40%-90% of new oropharyngeal cancer cases are HPV-associated [77], this lack of knowledge is concerning.

**Table 16. Students discuss how fear can drive their decisions around HPV and HPV vaccination.**

'I remember when all the women down south's results came back different or wrong, and that really scared me. I think that was only like two or three years ago. And I didn't know what cervical screening was until then.' (S10, Focus group 2; Female only Catholic school)

'My aunt had cervical cancer the year that I got that. And that scared me. So I was like, OK, I want that.' (S10, group 2; Female only Catholic school)

'I think you should scare them a wee bit, but not to put them off for life. So I think you should give them that wee scare.' (S33, Focus group 7; Male only Catholic school)

'I don't like needles. I didn't want to get it [the HPV vaccination].' (S8, Focus group 2; Female only Catholic school)

'I feel like some people are so scared of injections.' (S19, Focus group 3; Female only Catholic school)

'You'd be nervous going. . .you'd have to sit there, open your legs [during screening]! (S16, Focus group 3; Female only Catholic school)

All students were strongly supportive of the addition of HPV education in school for their age group (15–16 years old). They agreed that, at this age, they were mature enough to receive in-depth information regarding all aspects of HPV including transmission routes, protection against HPV and management of HPV. Students were divided on whether comprehensive information should be introduced at 14 year old. Those who promoted this education felt that this was very important as some students are sexually active at 14 years old and they feel that this is the right age to be informed prior to becoming sexually active. Those who supported comprehensive information at 15–16 years old were concerned that providing this information at 14 years old would be too scary. Interestingly, a recent study in the UK of 11,000 adolescents indicated that only 3.2% of 14 year olds had engaged in oral sex or full sexual intercourse [78]; this is in contrast to earlier studies which indicates that 20–30% of 14 years olds were engaging in higher risk sexual behaviours in previous decades [78, 79].

Several aspects regarding the format of the HPV education were considered important by students in order to optimise attention and improve memory of the education. One aspect discussed included whether non-scientific terminology should be used during this education to support and clarify sexual terms like oral sex. Males indicted that using non-scientific terminology would reduce the professionalism of the teaching while females felt that including non-scientific terminology would increase their understanding of the topics covered. Interestingly, this difference in female and male interpretation of sexual behaviour terms is not new and has been widely explored in previous publications [80, 81]. In the past, researchers have reported that males and females assign different meaning to sexual terminology with males generally knowing a greater range of non-scientific expressions related to sexual activity than females [80]. While gender differences between perceptions regarding sexual behaviour terminology may exist, the majority of students indicated that HPV education should be taught with students of all genders mixed together. While most students from the girls' Catholic school felt that separating genders was antiquated, many of them indicated that they would still be embarrassed and reluctant to discuss sexual health with males present. This finding, in alignment with previous research, suggests that single gender schools may predispose students to experience mixed-gender anxiety, potentially having a negative effect on future social interactions with the opposite sex [82].

To optimise interaction and attainment of HPV information, all students indicated that small groups of 6–10 students would be ideal. While this may pose practical challenges, this thinking is very consistent with evidence, which supports group sizes of 5–10 for optimal interaction within the group [83]. While some researchers have indicated that familiarity in a focus group can lead to peer-pressure and under-disclosure [83], students in this study consistently felt that being in the same group as their friends for this education, would create a safe environment and provide them with the opportunity to express themselves freely.

Students felt that a variety of interactive teaching methods should be utilised in a short HPV education session to maximise participation and retention of information. Social constructivist theories stress the importance of interactive teaching and its effect on focusing students' attention and increasing motivation [84]. While female students were in favour of the use of personal stories through videos to aid decision-making regarding HPV, males were less likely to view these videos as a learning tool and anticipated that males would not take the videos seriously and ridicule and disrespect the speaker in the video. Males in this study also indicated that the facilitator of the HPV education session should have teaching experience in order to 'control' the more immature males in the class. A recent study 2017 conducted in post-primary schools in Northern Ireland supported this finding [85], indicating that male stereotypes are constantly reinforced by educators and their peers contributing to boys becoming more disruptive than girls in class.

Students felt that they are competent at 16 years of age to make their own decisions about protecting themselves from HPV and should be able to self-consent to all vaccinations including HPV vaccinations. While students in this study expressed that they would want their parents to be involved in their HPV vaccination decision-making, they ultimately felt that they should make the decision. Mixed views were expressed regarding whether 15-year-old students would have the maturity to self-consent with males generally expressing that not all students would be mature at that age. Students desired more information regarding their legal rights to self-consent. They commented that inability to self-consent limits their participation in research. A systematic review by Fisher et al. (2019) [86] revealed that the main barrier to adolescent self-consent was fear from healthcare providers and teachers regarding parental response, even where students were deemed competent to make their own decisions. In parts of the UK, some healthcare professionals assess competency for vaccination using a Gillick consent tool [87, 88]; however, IMNs are often confused about the legal rights of students [7, 9, 89], similar to students in this study. It is important to note that students 16 years and above and Gillick competent students under 16 years old, can legally consent to the HPV vaccination without their parents' consent in Northern Ireland [52].

**5.1.1 Actions needed to increase capability to support behaviour change.** To enable students to make sexual health decisions to protect themselves from acquirement of HPV, more information needs to be provided to them in the form of additional school-based HPV education. This HPV education should highlight the various types of cancers associated with HPV acquirement, especially those associated with males.

Given the potential differences in gender perceptions of sexual behaviour, the HPV education should be approached sensitively in a professional manner with limited verbal use of non-scientific terminology but clear explanations provided of all terminology used. Understanding of content could be gaged through the use of an anonymised audience response system (ARS) during the HPV education [90]. ARSs have been shown to capture students' attention and increase concentration and interactions [84, 90].

Introducing a range of teaching methods during the HPV education for this age group is important. In developing a HPV intervention, it is important that facilitators undergo gender equality, diversity and inclusion training and a minimum level of teacher training in order to optimise the teaching experience for all students.

Students, educators and nurses delivering HPV vaccines, need to have a strong knowledge of the rights of students to self-consent to HPV vaccination. This should be implemented into the HPV education provided for this age group. Gottvall et al. (2015) [91] suggest that a relational approach is needed to promote dialogue between all parties with special care for the relationships between students and their parents.

## 5.2 Opportunity

Students indicated that they have not been provided with sufficient social and physical opportunities to receive education regarding HPV and HPV vaccination.

They felt that, while parents were supportive of their learning, parents were often uneducated and made decisions related to HPV vaccination based on their experiences of social media platforms or relatives' experiences. Consistent with these findings, a recent post-COVID-19 study found that two of the three strongest factors influencing whether a parent would make a change to vaccinate their child included the parents' vaccination history and their perception of the child's risk [92]. In a recent UK survey of 186 parents [93], only half of the parents had heard of HPV prior to completing the survey. Knowledge related to HPV-associated conditions in males was particularly low.

This finding of low parental HPV knowledge aligns with findings from other countries [94, 95]. However, Sherman & Nailer found that even providing brief information regarding HPV, persuaded the majority of parents of the importance of HPV vaccination in all genders [93]. Sherman & Nailer highlight an urgent need for public HPV education, particularly highlighting relevance to males, in order to facilitate educated decision-making. Students who were from single sex schools affiliated with the Catholic religion felt that their sexual health education was negatively impacted by this affiliation. This finding aligns with previous research findings that indicate that sexuality education in Catholic schools focuses on abstinence with little to no education regarding safe sexual practices and STIs [96, 97]. Consequently, students felt that HPV education should be a mandated component of a larger sexual health education programme to ensure that students from all religions, cultures and backgrounds are receiving this information in a consistent manner in post-primary school.

Studies from various countries indicate that young people with higher rates of religious attendance, have poorer sexual health knowledge [98, 99] with religious teachings often emphasising values like abstinence rather than teaching facts related to sexual health [37, 75, 100]. Schools inconsistently enlisted 'Love for Life' to provide some STI education and when this education was implemented consistently each year in a school, the students rated this highly though they indicated that the level of detail could be enhanced. While students found facilitation from staff from 'Love for Life' open and non-judgemental, 'Love for Life' is a Christian charity with their own principals including prioritising delaying sex [47] so their approach may reflect this aim.

Students felt that it was essential that HPV education at 15–16 years old should be followed with an opportunity to receive the HPV vaccine in school. They indicated that barriers to obtaining the HPV vaccine outside of school would include having to communicate with their parents about why they want the vaccine in order to get transport to their general practice. For most parents and their children, talking about topics related to sexuality creates anxiety and apprehension, especially if parents disapprove of their child's sexual choices [101].

**5.2.1 Actions needed to increase opportunity to support behaviour change.** Public campaigns are needed to highlight the potential impact of HPV, especially to males. Currently throughout the UK, parents are provided with HPV education in the form of a letter and standardised NHS information leaflet [102–105]. However, parental HPV education should be provided with a greater range of educational resources through the schools, with multilevel intervention approaches [106] combining social media platform resources [107, 108], interactional videos and other resources [92] grounded in behavioural theory [53]. This education should teach parents of the positive implications of parent-child communication regarding sexual health topics including more consistent contraceptive use [101]. HPV education should be implemented into a mandated sexual health programme in post-primary schools as part of the curriculum to ensure equal access to this education.

Churches in Northern Ireland, particularly of Catholic faith, need to be supported in deepening their understanding of the importance of HPV vaccination promotion in their communities, remaining faithful to the church's vision but recognising the contemporary context of life today. The Education Authority in conjunction with the Department of Health could play an important role in implementing HPV vaccine promotion strategies within faith-based communities to promote safe and open dialogue for health communication messages to be disseminated in a familiar and trusted setting [109].

### 5.3 Motivation

Students were motivated to protect themselves against HPV as they recognised that they could be potentially at risk of acquiring HPV if they were becoming sexually active. They described being positively motivated by media celebrity stories but recognised that social media can provide misleading and sometimes negative views of HPV and cervical screening. The majority of students in this age group use multiple social media platforms like Instagram and Snapchat and these platforms are an important source of their news [110]. In a 2016 study, compared to 83% of adults, only 44% of teenagers agreed that they could tell the difference between fake and real news [111]. In the past, celebrities with a cancer diagnosis have influenced young people to attend health checks, research cancers in more depth and possibly even contributed to change in health policy [112].

Students expressed being motivated by fear, which was driven by social media and/or their own personal experiences of HPV associated cancers in their social circle. Some students felt that scaring students was necessary in the HPV education to motivate them, though these students indicated that scaring students 14 years or under would not be appropriate. Studies suggest that fear can be effective in increasing vaccination uptake [113, 114] but warn that fear appeal in public campaigns may be perceived by adolescents as controlling them into message conformity [114]. Some students expressed fear of needles as being a barrier to student motivation to receive the HPV vaccines though indicated that this was higher at 12–13 years old than at 15–16 years old.

Students felt that having more information about HPV would encourage safer sexual behaviour and encourage attendance at HPV screening offered to them in the future. The males in the group, none of which were vaccinated, indicated that they would take the HPV vaccine if offered to them in school. This is consistent with studies which show that even providing short and limited HPV education, can increase intention to be vaccinated [53].

Students expressed that student embarrassment could hinder communication during HPV education and emphasised the importance of the facilitator's characteristics in order to create a motivational environment. Openness and being non-judgemental were deemed the most important characteristics of a facilitator for delivery of HPV education alongside expertise in this area. Many students felt that having younger, more socially current facilitators around 25–30 years old would be ideal though age was rated as being much less important that the openness and expertise of the facilitator. They felt that it would be less embarrassing to have this subject taught by a person that they did not know, external to the school. Students commonly report embarrassment, shame and judgement as barriers to conducting open discussions about sexual health matters [115, 116] and therefore this is an extremely important consideration in the development of any HPV education for this age group. Given this information, all students identified that the IMNs would be the most appropriate professionals to deliver this education alongside delivery of the vaccination, recognising their expertise in this area. Being external to the school, students felt that they would be able to retain anonymity making it easier for them to ask questions and participate in discussions on the topic. Only one group of students identified the school nurse as a potential facilitator of this education, despite three out of four schools involved having an allocated school nurse. Despite being highly qualified, both educators and parents often perceive the role of the school nurse to be purely administration of medication and first aid with little recognition for their potential role in health promotion [117, 118]. Therefore, given their cultural environment, it is logical that students would also undervalue the skill set of school nurses as was evident in the study. IMNS and school nurses both express a strong desire to be more directly involved in health promotion and indicate that, historically, they were provided with time and resources for this purpose [9, 55].

However, both IMNs and school nurses express a lack of support in terms of time and resources allocated currently to develop roles in these areas of health education [9, 55]. Students felt that, apart from some science teachers, teachers would not have the topic expertise to teach HPV education and that embarrassment would hinder open discussions with most of their teachers. This finding is consistent with teachers' opinions as those with no specific post-graduation public health education, also expressed a lack of knowledge and desire to teach sexual health education [119, 120].

**5.3.1 Actions needed to increase motivation to support behaviour change.** Open-minded and non-judgemental nurses, with a public health qualification and phlebotomy qualification, should provide HPV education as external partners through post-primary school in order to increase student motivation to engage in this education. Fake news social media platforms should be discussed with students along with providing standardised evidence-based teaching about HPV-associated issues. The introduction of celebrity personalities may positively influence motivation for protecting against HPV though studies suggest that for identification to occur, the two parties in a para-social relationship must share some characteristics [112]. Public campaigns should balance health promotion messages carefully and avoid overly strong fear appeal messages. From April 2022, the World Health Organisation (WHO) concluded that either one or two HPV injections only are needed for 15 year olds instead of the previously administered three injections [121]. This could benefit HPV uptake for those students to have a strong aversion to needles.

## 6. Strengths and limitations

One of the strengths of this study is the use of the COM-B model and TDF domains to identify the aspects which need to change to enable students to protect themselves against acquirement of HPV. The National Institute for Health and Care Excellence (NICE) supports the use of this model when designing sexual health interventions in the UK [122]. However, due to the nature of focus groups, only 10 of the 14 TDF domains were targeted so some additional information may potentially have been missed through using this format.

While the study includes students with different genders and religions, the majority of students in the four schools have a middle to higher socioeconomic background so views could vary in schools with a higher portion of students from lower socioeconomic backgrounds. Additionally, the study included a higher percentage of females compared to males, which may have impacted the findings.

It is likely that participating schools are already more supportive of the HPV vaccination programme than schools which decided not to participate in the study which may impact the results. Teachers also may have targeted students who spoke more articulately and were more mature than the average student in their class. Due to the lack of ethic diversity and unique religious history in Northern Ireland, the results may not be generalisable to students in the rest of the UK.

## 7. Conclusion

Students, 15–16 years old, felt that they were mature enough to self-consent to HPV vaccination and make decisions regarding their sexual health. However, they indicated that they lacked sufficient HPV education and education regarding their legal right to consent to vaccination, to empower them to make these decisions. They supported the introduction of mandatory age-appropriate HPV education in all post-primary school year groups to provide consistent HPV content and delivery. They indicated that face-to-face interactive HPV education could easily be incorporated into their curriculum and should be delivered by an external

expertise who is open and approachable. Such changes would need to be supported by the Education Authority in conjunction with the Department of Health. Students felt that removal of these barriers would lead to safer sexual practices, increased awareness of the importance of HPV screening and increased HPV vaccination uptake.

## Supporting information

**S1 Appendix. Focus group questions asked to participants in alignment with the COM-B model and TDF domains.**
(DOCX)

## Acknowledgments

We would like to express our sincere gratitude to the teachers and students who organised and participated in this study.

## Author Contributions

**Conceptualization:** Terri Flood, Marian McLaughlin.

**Data curation:** Terri Flood.

**Formal analysis:** Terri Flood, Iseult Wilson, Marian McLaughlin.

**Funding acquisition:** Terri Flood, Marian McLaughlin.

**Investigation:** Terri Flood, Marian McLaughlin.

**Methodology:** Terri Flood, Marian McLaughlin.

**Project administration:** Terri Flood, Ciara M. Hughes, Iseult Wilson.

**Resources:** Terri Flood.

**Software:** Terri Flood.

**Supervision:** Terri Flood, Ciara M. Hughes, Iseult Wilson, Marian McLaughlin.

**Validation:** Terri Flood.

**Visualization:** Terri Flood.

**Writing – original draft:** Terri Flood, Marian McLaughlin.

**Writing – review & editing:** Terri Flood, Ciara M. Hughes, Iseult Wilson, Marian McLaughlin.

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
