## [Decision Letter · Decision Letter 0]

22 Jan 2024

PGPH-D-23-01569

Applying the COM-B behaviour model to understand factors which impact 15-16 year old students’ ability to protect themselves against acquirement of Human Papilloma virus (HPV) in Northern Ireland, UK.

Dear Dr. Flood,

Thank you for submitting your manuscript to PLOS Global Public Health. After careful consideration, we feel that it has merit but does not fully meet PLOS Global Public Health’s publication criteria as it currently stands. Therefore, we invite you to submit a revised version of the manuscript that addresses the points raised during the review process.

Please see the comments of two reviewers below and in the attached document. Please note that you may rebut any of the reviewer comments that you do not feel need a corresponding revision. For instance, where the reviewers have suggested amending the conclusions, I believe that those sentences can be retained.

We look forward to receiving your revised manuscript.

Kind regards,

Hanna Landenmark

Staff Editor

Journal Requirements:

Additional Editor Comments (if provided):

Reviewers' comments:

Reviewer's Responses to Questions

**Comments to the Author**

1. Does this manuscript meet PLOS Global Public Health’s publication criteria? Is the manuscript technically sound, and do the data support the conclusions? The manuscript must describe methodologically and ethically rigorous research with conclusions that are appropriately drawn based on the data presented.

Reviewer #1: Yes

Reviewer #2: Yes

2. Has the statistical analysis been performed appropriately and rigorously?

Reviewer #1: Yes

Reviewer #2: Yes

3. Have the authors made all data underlying the findings in their manuscript fully available (please refer to the Data Availability Statement at the start of the manuscript PDF file)?

Reviewer #1: Yes

Reviewer #2: Yes

4. Is the manuscript presented in an intelligible fashion and written in standard English?

Reviewer #1: Yes

Reviewer #2: Yes

5. Review Comments to the Author

Reviewer #1: 1- The role of the church in sexual education seems to be absent from this article. It would be beneficial to include some background information on this aspect, as one would anticipate that both parents and students in such religious settings would be influenced by the church’s stance.

2- Could you provide additional details on the planning and execution of the focus groups? Was there consistency in group composition across sessions, or did the same participants engage in every discussion within their respective groups? In certain groups, you have exclusively Catholic participants, while others comprise a mix of religious beliefs. Do you believe that this variation could influence the outcomes of your study?

I miss the role of church within sex education

3-Could the church play a role in raising awareness about this issue? It might be worthwhile to consider adding this perspective to the “Actions Needed to Support Behavior Change” section of the article. This could provide a comprehensive view on how religious institutions can contribute to the discourse and promote positive change.

Reviewer #2: Dear author, The comments and additional clarifications needed are notified with yellow highlights and in comments box for corrections. The manuscript with review comments is herewith attached for reference.

6. PLOS authors have the option to publish the peer review history of their article (what does this mean?). If published, this will include your full peer review and any attached files.

**Do you want your identity to be public for this peer review?** For information about this choice, including consent withdrawal, please see our Privacy Policy.

Reviewer #1: **Yes: **Dr. Amani Eltayb

Reviewer #2: **Yes: **Abdul Nazer Ali

---

## [Decision Letter · Decision Letter 1]

7 Mar 2024

PGPH-D-23-01569R1

Applying the COM-B behaviour model to understand factors which impact 15-16 year old students’ ability to protect themselves against acquirement of Human Papilloma virus (HPV) in Northern Ireland, UK.

Dear Dr. Flood,

Thank you for submitting your manuscript to PLOS Global Public Health. After careful consideration, we feel that it has merit but does not fully meet PLOS Global Public Health’s publication criteria as it currently stands. Therefore, we invite you to submit a revised version of the manuscript that addresses the points raised during the review process.

We have received minor comments from reviewer 2, particularly regarding the references. Please take this opportunity to proofread the manuscript, as there may not be another chance for you to do so after the acceptance.

We look forward to receiving your revised manuscript.

Kind regards,

Siyan Yi, MD, MHSc, PhD

Academic Editor

Journal Requirements:

2. Please provide separate figure files in .tif or .eps format only and remove any figures embedded in your manuscript file. Please also ensure all files are under our size limit of 10MB.

Additional Editor Comments (if provided):

Reviewers' comments:

Reviewer's Responses to Questions

**Comments to the Author**

1. If the authors have adequately addressed your comments raised in a previous round of review and you feel that this manuscript is now acceptable for publication, you may indicate that here to bypass the “Comments to the Author” section, enter your conflict of interest statement in the “Confidential to Editor” section, and submit your "Accept" recommendation.

Reviewer #2: All comments have been addressed

2. Does this manuscript meet PLOS Global Public Health’s publication criteria? Is the manuscript technically sound, and do the data support the conclusions? The manuscript must describe methodologically and ethically rigorous research with conclusions that are appropriately drawn based on the data presented.

Reviewer #2: Yes

3. Has the statistical analysis been performed appropriately and rigorously?

Reviewer #2: Yes

4. Have the authors made all data underlying the findings in their manuscript fully available (please refer to the Data Availability Statement at the start of the manuscript PDF file)?

Reviewer #2: Yes

5. Is the manuscript presented in an intelligible fashion and written in standard English?

Reviewer #2: Yes

6. Review Comments to the Author

Reviewer #2: The references section needs correction with uniform referencing style.

7. PLOS authors have the option to publish the peer review history of their article (what does this mean?). If published, this will include your full peer review and any attached files.

**Do you want your identity to be public for this peer review?** For information about this choice, including consent withdrawal, please see our Privacy Policy.

Reviewer #2: **Yes: **Abdul Nazer Ali

---

## [Editor Report · Decision Letter 2]

21 Mar 2024

Applying the COM-B behaviour model to understand factors which impact 15-16 year old students’ ability to protect themselves against acquirement of Human Papilloma virus (HPV) in Northern Ireland, UK.

PGPH-D-23-01569R2

Dear Ms Flood,

We are pleased to inform you that your manuscript 'Applying the COM-B behaviour model to understand factors which impact 15-16 year old students’ ability to protect themselves against acquirement of Human Papilloma virus (HPV) in Northern Ireland, UK.' has been provisionally accepted for publication in PLOS Global Public Health.

Best regards,

Siyan Yi, MD, MHSc, PhD

Academic Editor